# Mechanisms Affecting the Acquisition, Persistence and Transmission of *Francisella tularensis* in Ticks

**DOI:** 10.3390/microorganisms8111639

**Published:** 2020-10-23

**Authors:** Brenden G. Tully, Jason F. Huntley

**Affiliations:** Department of Medical Microbiology and Immunology, University of Toledo College of Medicine and Life Sciences, Toledo, OH 43614, USA; Brenden.Tully@rockets.utoledo.edu

**Keywords:** *Francisella tularensis*, tularemia, tick-borne disease

## Abstract

Over 600,000 vector-borne disease cases were reported in the United States (U.S.) in the past 13 years, of which more than three-quarters were tick-borne diseases. Although Lyme disease accounts for the majority of tick-borne disease cases in the U.S., tularemia cases have been increasing over the past decade, with >220 cases reported yearly. However, when comparing *Borrelia burgdorferi* (causative agent of Lyme disease) and *Francisella tularensis* (causative agent of tularemia), the low infectious dose (<10 bacteria), high morbidity and mortality rates, and potential transmission of tularemia by multiple tick vectors have raised national concerns about future tularemia outbreaks. Despite these concerns, little is known about how *F. tularensis* is acquired by, persists in, or is transmitted by ticks. Moreover, the role of one or more tick vectors in transmitting *F. tularensis* to humans remains a major question. Finally, virtually no studies have examined how *F. tularensis* adapts to life in the tick (vs. the mammalian host), how tick endosymbionts affect *F. tularensis* infections, or whether other factors (e.g., tick immunity) impact the ability of *F. tularensis* to infect ticks. This review will assess our current understanding of each of these issues and will offer a framework for future studies, which could help us better understand tularemia and other tick-borne diseases.

## 1. Introduction

The most recent data from the United States (U.S.) Centers for Disease Control and Prevention (CDC) indicate that the reported number of tick-borne disease cases have more than doubled in the past 13 years and account for 77% of all vector-borne disease cases [1]. Although insecticides historically have been useful for controlling tick-borne diseases, resistance is becoming widespread in the U.S. [1,2]. Tick-borne diseases, including Lyme disease, spotted fever rickettsioses, babesiosis, anaplasmosis/ehrlichiosis, and tularemia, have been difficult to control because vaccines are not available. Additionally, reports of tick geographic range expansion and increases in wildlife populations that support ticks further complicate tick-borne disease control efforts [3,4,5]. Although Lyme disease accounted for 82% of all U.S. tick-borne disease cases between 2004 and 2016, tularemia cases have seen a resurgence in recent decades. Over 225 tularemia cases per year have been reported since 2015, with 314 reported cases in 2015—the most recorded since 1964 [1].

*Francisella tularensis*, the causative agent of tularemia, has been classified as a Tier 1 Select Agent by the CDC because of its low infectious dose, ease of aerosolization, rapid onset of severe disease, and high morbidity and mortality rates. Two *F. tularensis* subspecies cause human disease and, although they are closely related genetically, vary in their infectious dose and disease severity [6]. *F. tularensis* subsp. *tularensis* (Type A) has an extremely low infectious dose (<10 CFU) and is associated with severe, often lethal, disease. *F. tularensis* subsp. *holarctica* (Type B), has a slightly higher infectious dose (>100 CFU) and is associated with progressive disease with lower mortality rates [7]. Type A *F. tularensis* is further divided into three distinct clades, A1a, A1b and A2. Previous studies have shown that mice infected with A1b die earlier than those infected with A1a, A2 or Type B *F. tularensis* strains, demonstrating marked differences in *F. tularensis* virulence [6]. Through repeated subculturing of *F. tularensis* subsp. *holarctica*, a live attenuated strain was created by the former Soviet Union in the 1930s and has been designated as the live vaccine strain (LVS) [8]. Despite its name, LVS is not a licensed vaccine in the U.S. due to the unresolved questions about the mechanism(s) of attenuation, adverse effects in some immunized humans, and incomplete protection against Type A aerosol infection [7,8]. However, LVS has been proven to be a useful tool to study *F. tularensis* virulence, as it causes lethal disease in mice yet can be worked with using normal biosafety precautions (i.e., biosafety level 2; BSL2) [9]. A separate *Francisella* species, *F. novicida*, rarely associated with disease in immunocompromised humans, is used as a surrogate for *F. tularensis* in some studies because of reduced biosafety requirements and ease of genetic manipulation [10].

Beyond concerns over the potential use of *F. tularensis* as a bioweapon (i.e., Select Agent designation), approximately half of tularemia cases in the U.S. are associated with tick bites [11,12]. In contrast, European tularemia cases are generally associated with ingestion of contaminated water from wells, streams, rivers, ponds, and lakes [13,14]. However, in Sweden, most tularemia cases are associated with mosquito bites [15,16,17]. A recently-published model indicated that climate change may triple the number of European tularemia cases per year, due to increases in mosquitos, higher temperatures, and increased precipitation [18]. Although ticks are generally not considered to be major drivers of European tularemia infections, *F. tularensis*-infected ticks have been reported in Spain [19], Germany [20], Denmark [21] and Poland [22,23]. Conversely, other groups have not detected *F. tularensis* in ticks collected from Poland, France, or the Netherlands [24,25,26], suggesting that more studies are needed to understand the current and future risks of tick-transmitted tularemia in Europe.

In the U.S., where tularemia was first documented, field-collected *Dermacentor andersoni* ticks (Rocky Mountain wood tick) were shown to transmit lethal *Bacterium tularense* (now known as *F. tularensis*) to guinea pigs [27]. However, the true role of *D. andersoni* ticks in U.S. tularemia cases remains unknown. Data from the CDC indicate that U.S. tularemia infections more commonly stem from *D. variabilis* (American dog tick) and *Amblyomma americanum* (Lone star tick) ticks, which are known to vary in their geographic distribution and mammalian hosts, as well as less understood factors, including tick physiology, endosymbionts, and antimicrobial defenses [4,28]. Indeed, previous studies have reported that differences in the numbers of tick phagocytic cells and prevalence/type of endosymbionts in *A. americanum*, *D. andersoni*, and *D. variabilis* ticks affected the molting success (i.e., survival between tick life stages) of these three tick species [29,30,31].

The complex life cycle of the tick (3 year progression from larvae to nymph to adult), including taking a blood meal from various hosts and molting to the next life stage after each blood meal, combined with varying lengths and severities of North American winters, indicates that upon infecting a tick, *F. tularensis* must undergo major changes over the course of >5 months to persist and replicate, before being transmitted to naive mammals. Laboratory experiments have confirmed that *F. tularensis* persists in ticks for >4 months, supporting the role of the tick as a potential environmental reservoir [32,33]. Additionally, *F. tularensis* has been shown to persist in ticks between molts (transstadial transmission) and replicate to high bacterial numbers in ticks, demonstrating that ticks serve as both a reservoir and an amplification vessel for *F. tularensis* in the environment [33,34]. Rabbits also have been implicated as a major environmental reservoir for tularemia, as studies have demonstrated that they can survive for 3–13 days following intradermal infection (mimicking a tick infection) with Type A1a, A1b or A2 *F. tularensis* and over 14 days for Type B *F. tularensis* infection [35]. This rabbit infection data, together with the 3–7 days that both *D. variabilis* and *A. americanum* ticks (varies depending on the tick life stage) take a blood meal, suggest that infected rabbits can potentially infect large numbers of ticks [4,33,34]. Despite those studies, we still do not understand whether different tick species acquire different levels of *F. tularensis* infections from different hosts, whether different tick species promote or restrict *F. tularensis* infections, what *F. tularensis*—tick interactions occur, how *F. tularensis* is maintained through the tick molt, or how infected ticks transmit *F. tularensis* to naïve hosts.

This review will highlight what is known about U.S. tick species that are associated with tularemia, factors that appear to promote *F. tularensis* infections in ticks, and how *F. tularensis* infections are maintained in ticks. This review also will highlight research areas where new or additional research studies are needed, including studies to elucidate *F. tularensis*—tick interactions, primary tick vectors for *F. tularensis*, mechanisms influencing *F. tularensis* infections of ticks, the ability of *F. tularensis* to persist in ticks, and mechanisms that promote *F. tularensis* transmission by ticks.

## 2. Epidemiology of *F. tularensis* Transmission by Ticks in the U.S.

Tularemia has been shown to be transmitted by at least four different ticks in the U.S., including *D. variabilis*, *D. andersoni*, *D. occidentalis* (Pacific coast tick), and *A. americanum* [27,33,34,36]. Although *D. variabilis* has been implicated as the primary vector, and *A. americanum* also appears to be an important vector for tularemia, we still do not know which tick(s) poses the greatest threat to human health in the U.S. [33,37]. A 1924 study noted that *D. andersoni* ticks could transmit virulent *F. tularensis* to guinea pigs [27] and a more recent study reported that *D. andersoni* ticks could be infected by and transmit *F. novicida* (rare infection of immunocompromised humans) [38]. However, very little is known about current infection rates of *D. variabilis*, *A. americanum*, or *D. andersoni* ticks with virulent strains of *F. tularensis* or about which ticks are commonly associated with current human tularemia cases. These, and other gaps in knowledge, have resulted in a call for new research studies on tick-borne *F. tularensis* [28]. In addition, although very few studies have explored *F. tularensis*–tick interactions, such studies could provide important information that could be used to develop new strategies to reduce *F. tularensis* in the environment [12].

One area of the U.S. that has been extensively studied to understand *F. tularensis* environmental persistence and tick transmission is Martha’s Vineyard. In both 1978 and 2000, two separate tularemia outbreaks occurred on Martha’s Vineyard, each involving 15 patients, with 1 fatality [39,40]. Although pneumonic disease was the most common symptom in both outbreaks, *D. variabilis* tick bites remain the only proven mode of transmission in most cases [41]. Sampling studies at Martha’s Vineyard have assessed *F. tularensis* infection rates in *D. variabilis* ticks, finding a median annual prevalence of 3.4% over four years, suggesting that *F. tularensis* infections are stable on the island [42]. Additionally, *D. variabilis* ticks infected with Type A *F. tularensis* have been reported to harbor over 10^8^ genome equivalents/tick. Although genome equivalents may not accurately quantitate viable bacteria, those data indicate that *D. variabilis* ticks may sustain high *F. tularensis* numbers that cause significant transmission and disease in humans [37,43].

The majority of tularemia cases occur in the south-central U.S. [33,42,44]. In fact, four states accounted for 58% of tularemia cases in 2018 (most recent data from the CDC): Arkansas (24%), Oklahoma (19%), Kansas (8%) and Missouri (7%) [45]. In this region of the U.S., two major tick species exist: *D. variabilis* and *A. americanum* [46,47]. In one study from Missouri, >8500 *A. americanum* ticks were harvested over three years from various hosts, including white-tailed deer, fox, opossum and rabbits, with tick prevalence rates on these potential hosts ranging from 0.7% to 100%. Although the presence and absence of pathogenic bacteria were not assessed in these *A. americanum* ticks, >1100 *A. americanum* nymphs were collected from a single rabbit, a natural reservoir for tularemia, highlighting the ability of one infected host to spread tularemia to thousands of other hosts, including humans [48]. In Arkansas, approx. 92% of field-collected ticks were *A. americanum* and approx. 7% were *D. variabilis*. Interestingly, none of the *D. variabilis* ticks tested positive for *F. tularensis* (>2000 ticks tested), whereas approx. 4% of *A. americanum* ticks (>5000 ticks tested) were positive for *F. tularensis* [49]. A study of over 3500 field-collected *D. variabilis* ticks from Minnesota identified Type A *F. tularensis* in 3.6% of those ticks [50]. That infection rate was similar to what has been reported at Martha’s Vineyard, where the annual *F. tularensis* infection rates in *D. variabilis* ticks ranges from 2.7% to 4.3%, demonstrating that tick-borne tularemia infections are not restricted to the south-central U.S. [42]. In contrast, sampling studies in Washington state did not identify *F. tularensis* in either *D. andersoni* or *D. variabilis* ticks. However, that study examined less than 200 *Dermacentor* sp. ticks and only 25 tularemia cases were reported in Washington state between 2011 and 2016 [51]. Clearly, more studies are needed to assess *F. tularensis* infection rates in multiple tick vectors across the U.S. Further, given the geographic range expansion of various ticks throughout the U.S. [4], continuing studies will be needed to understand how this expansion will affect tick-borne disease transmission.

Although this review is focused on *F. tularensis* infections of ticks, historical data suggest that biting flies also can transmit tularemia [52,53]. However, no recent data from the CDC have linked tularemia infections with biting flies. One study used *Drosophila* as an arthropod model for *F. tularensis* infections, finding that doses as low as 200 CFU killed >90% of fruit flies injected with *F. tularensis* LVS [54], bringing into question whether flies play a significant role in tularemia transmission.

## 3. Factors that Affect *F. tularensis* Infections of Ticks

Tick-borne pathogens must be able to efficiently transition from mammalian to arthropod hosts following tick feeding [55,56]. *F. tularensis* has been reported to infect and cause disease in over 300 animal species, including humans, highlighting the zoonotic potential and plasticity of *F. tularensis* [57]. Although not well studied, *F. tularensis* likely undergoes substantial changes, including major changes in protein expression profiles, between the mammalian host and the tick vector. Factors such as temperature and pH have been shown to be important cues when *Borrelia burgdorferi*, the causative agent of Lyme disease, transitions between mammalian hosts and ticks. These stimuli result in modifications of bacterial surface proteins to enhance *B. burgdorferi* acquisition by ticks [58,59]. Two surface-exposed lipoproteins, OspA and OspC, are among the most well-characterized proteins that are differentially expressed in *B. burgdorferi*. OspA is highly expressed under conditions that resemble the tick environment (pH 7.5 and 23 °C) [60]. Conversely, during a tick blood meal, the pH and temperature of the tick midgut change to 6.8 and 35 °C, respectively, triggering the upregulation of OspC, and promoting the migration of *B. burgdorferi* through the tick salivary gland to the mammalian host [61,62].

In mammals, *F. tularensis* is an intracellular pathogen, infecting cell types ranging from macrophages, to neutrophils, to epithelial cells, to erythrocytes [63]. Many previous studies have identified *F. tularensis* virulence factors and have examined *F. tularensis* pathogenesis mechanisms using macrophage infection models [64]. When comparing macrophages and ticks, *F. tularensis* encounters low pH in both the macrophage phagosome and tick midgut [62,65]. In fact, both Type A *F. tularensis* and LVS are resistant to acid stress and viable at pH 3 [66]. While it has been reported that *F. tularensis* responds to low pH by upregulating genes in the *Francisella* pathogenicity island (FPI) to escape from the phagosome [67,68], we are not aware of any study that directly examines whether *F. tularensis* uses low pH as an indicator of the transition between mammalian and arthropod hosts.

Conversely, because iron is extremely limited in the macrophage phagosome but is readily available in replete ticks (through hemolysis), it is possible that *F. tularensis* may successfully transition from mammalian to arthropod hosts by sensing changes in iron and/or altering expression of iron-regulated genes in the tick (Figure 1) [69]. Iron-regulated genes have been shown to be important for the regulation of virulence in *B. burgdorferi* [70]. One study found that *F. tularensis* LVS differentially regulated over 70 genes in iron-limiting conditions, many of which were shown to be associated with virulence or intracellular replication [71]. Although that study did not explore *F. tularensis* gene regulation under iron-replete or high-iron conditions, similar to the tick midgut after a blood meal, such data could provide important information about how *F. tularensis* initially responds and adapts to life inside a tick.

A third environmental cue that *F. tularensis* might use to successfully transition to ticks may be temperature changes (Figure 1). One study reported that 11% of the *F. tularensis* LVS genome was differentially regulated when the bacterium was switched from ambient (26 °C) to mammalian (37 °C) temperature. Up to 40% of those identified genes were known to be important for virulence or intracellular replication, suggesting that temperature changes prime *F. tularensis* for pathogenicity in mammals [72]. Although that study may have provided information relevant to *F. tularensis* transmission from ticks to mammals, no studies have been performed to identify *F. tularensis* genes differentially regulated when transitioning from mammals to ticks. Another study found that *F. tularensis* modifies its lipopolysaccharide (LPS) structure in response to temperature changes, including altering expression of acetyltransferases, which add shorter or longer acyl chains to lipid A under ambient (18 °C) or mammalian (37 °C) temperatures, respectively [73]. Those LPS modifications were shown to promote bacterial survival and growth in cold conditions, which could be speculated to help *F. tularensis* survive and persist in ticks during the winter. However, experiments to confirm whether these LPS modifications promote *F. tularensis* persistence in ticks have not been conducted.

Likely because of the difficulties and biosafety risks of working with infected ticks, tick cell lines offer a simplified tool to understand how variables, such as temperature and tick species, can affect bacterial infection and persistence in arthropods. In one study, cell lines derived from *D. andersoni* and *Ixodes scapularis* ticks were infected with *F. novicida*, finding that at 34 °C, *F. novicida* infected and replicated 2-logs higher in *D. andersoni*-derived cells, compared to *I. scapularis*-derived cells. However, *F. novicida* infection killed up to 25% of *D. andersoni* cells, compared to *I. scapularis* cells that appeared to be unaffected by *F. novicida* up to 6 days post-infection. At 24 °C, *F. novicida* infected and replicated to similar levels in *D. andersoni*-derived cells, compared to *I. scapularis*-derived cells. However, approximately 15% less *D. andersoni* cell death was detected at the lower temperature, despite elevated bacterial numbers, indicating that low temperatures may decrease bacterial virulence, while still supporting bacterial replication in tick cells [5]. Although the above highlighted studies provided important insights into how *F. tularensis* may sense environmental cues and promote tick infections, there are still major gaps in our understanding of how virulent *F. tularensis* strains infect ticks, how *F. tularensis* adapts to life in the tick, and which ticks pose the greatest risk for tularemia transmission.

## 4. *F. tularensis* Persistence and Transmission in Ticks

As highlighted above, *F. novicida* was better able to infect and persist in *D. andersoni*-derived cells, compared with *I. scapularis*-derived cells [5]. Lower numbers of *F. novicida* in *I. scapularis*-derived cells may have been due to a wide range of factors, including decreased availability of nutrients (i.e., nutritional immunity) or inability of *F. novicida* to evade tick defense mechanisms [5]. However, limitations of those studies included the short time frame examined (i.e., 120 h), use of a non-human pathogen, and use of cell lines that are unlikely to mimic the complex processes required for a tick infection. As is true for many tick-borne pathogens, future studies should carefully examine acquisition of the pathogen from an infected mammalian host to the tick vector, pathogen persistence through the molt and/or winter, migration of the pathogen through the tick salivary glands or regurgitation of the pathogen from the tick midgut, and pathogen infection of a mammalian host following a tick blood meal [61,74].

Although the tick blood meal provides a nutrient-rich environment for many pathogens, these resources are depleted/digested within a few weeks [75] (Figure 1). As a result, long-term nutrient acquisition by pathogens in ticks becomes a major restriction. The minimum growth requirements of *F. tularensis* are well known, including iron, cysteine, and at least 12 other amino acids [76]. However, it is unclear how *F. tularensis* acquires such nutrients in the tick, particularly after the blood meal has been digested. Another consideration is that the tick blood digestion process generates harmful intermediates/breakdown products that damage microbes [77]. Although *F. tularensis* has been reported to encode a number of enzymes that protect the bacterium from free radical/reactive oxygen species damage [78,79], the role of these enzymes in protecting *F. tularensis* in the tick remains unstudied.

The majority of tick-borne bacterial pathogens, including *Rickettsia rickettsii, Anaplasma phagocytophilum, Babesia microti,* and even *B. burgdorferi,* replicate intracellularly within the tick. By infecting tick cells, such as the midgut epithelium, bacteria may avoid tick antimicrobial responses (e.g., defensins, lysozymes, microplusins) [61,80,81] or toxic breakdown products from the tick blood meal [82,83]. With respect to *F. tularensis*, high bacterial numbers have been detected in *D. variabilis* midguts and hemolymph after taking a blood meal from mice, while limited bacterial numbers were detected in the tick salivary glands [34]. Avoiding the use of live vertebrate hosts, two separate studies utilized capillary tube feeding methods to infect either *D. variabilis* or *A. americanum* ticks with *F. tularensis* LVS. In those studies, *F. tularensis* LVS was detected in *D. variabilis* and *A. americanum* salivary glands and midguts. However, transmission to naïve hosts was not examined in either study [33,84]. While those studies indicated that both *D. variabilis* and *A. americanum* ticks could support *F. tularensis* infections as well as transstadial transmission [11,12], capillary feeding bypassed *F. tularensis* exposure to blood and immune cell molecules and failed to mimic numerous cues that *F. tularensis* likely encounters during the mammal to tick transition (Figure 1). Separate studies demonstrated that *D. variabilis* ticks could transmit Type A1b (strain MA00-2987), Type A2 (strain WY96-3418), and Type B (strain KY99-3387) *F. tularensis* infections with high efficiency (e.g., between 58% and 89% of naïve mice (*n* = 12–23) were infected) [37]. However, similar studies have not been performed using *A. americanum* ticks and the ability of virulent *F. tularensis* strains to infect and persist in these ticks is unknown.

## 5. *F. tularensis* Biofilms

Biofilm formation has been shown to enhance resistance to various stressors, including antimicrobials, hypoxia, and nutrient limitation [85,86,87]. In the case of *Vibrio cholerae,* biofilm formation on crab shells has been shown to be an important carbon source, contributing to bacterial persistence during the aquatic cycle [88]. Additionally, *Yersinia pestis* has been shown to form biofilms in fleas, which aids in persistence and transmission to mammalian hosts [89]. *F. tularensis* has been isolated from a range of environments, including arthropods [27,90,91]. Given that chitin is widely available in nature and is a major component of the tick cuticle [92], it has been speculated that chitin may promote *F. tularensis* biofilm formation and may serve as a carbon source for *F. tularensis* in ticks. However, the true role of biofilms in *F. tularensis* virulence and pathogenesis remains unclear (Figure 1) due to the fact that many biofilm studies have been performed using *F. novicida,* an opportunistic pathogen. Previous studies demonstrated that *F. novicida* rapidly forms robust biofilms on a variety of surfaces, including plastic, glass, and crab shells [93,94,95]. In contrast, although *F. tularensis* Type A and Type B strains also have been shown to form biofilms, those biofilms were less dense and formed at slower rates than *F. novicida* [93,96]. Given differences in *in vitro* biofilm formation among the three *Francisella* species/subspecies (*F. novicida*, Type A, and Type B) and unanswered questions about how *F. tularensis* persists in ticks, future studies should examine biofilm formation in ticks by virulent *F. tularensis* strains.

## 6. *F. tularensis* Chitin Utilization

One study reported that *F. novicida* requires two chitinases, *chiA* and *chiB*, part of the same metabolic pathway, to use chitin as a sole carbon source and forms biofilms on chitin (crab shells) [93]. Conversely, another group demonstrated that, compared to wild-type *F. novicida, chiA* and *chiB* mutants produced more robust biofilms in negatively-charged tissue culture plates and these thicker biofilms were more resistant to the charged aminoglycoside antibiotic gentamicin, but not ciprofloxacin. Those results lead the authors to speculate that chitinases also may function by changing the surface charge of *F. novicida*, allowing for changes in surface attachment and biofilm formation [97]. Although only one study has examined chitinase mutants in ticks, all three *F. novicida mutants* (*chiA*, *chiB,* and *chiAB*) were detected in *D. andersoni* tick midguts and salivary glands 4 days after feeding on infected mice. However, given the short time frame of those studies and noted variations in bacterial detection in tick samples, it still is unclear whether chitinase genes are required for virulent *F. tularensis* to infect and persist in ticks [38].

During the tick molting process (i.e., larvae to nymph; nymph to adult), the tick cuticle, of which chitin is a major component, is degraded, remodeled, and new components are synthesized [98]. Chitin is a polymer of N-acetylglucosamine (GlcNAc) and it has been speculated that GlcNAc, and other chitin components, may be released during tick molting. The peritrophic matrix (Figure 1), a semipermeable chitinous membrane that surrounds the tick blood meal, also has been speculated to breakdown and remodel during tick molting [98]. Although little is known about how *F. tularensis* persists in ticks through the molting process or whether *F. tularensis* can utilize free chitin fragments liberated during tick molting, one previous study reported that *F. tularensis* numbers dramatically increased in *D. variabilis* ticks following the nymph-adult molt [34].

## 7. Tick Endosymbionts

Recent studies on tick microbiomes have raised questions regarding how various endosymbionts (symbiotic and usually non-pathogenic microbes) impact the ability of ticks to harbor pathogens [99]. Given that tick microbiota diversity appears to vary based on many factors, the acquisition, persistence, and transmission of tick-borne diseases may be directly impacted by differences in tick microbiota [31,99]. In addition to altering tick immune responses, bacterial endosymbionts have been shown to provide beneficial vitamins and nutrients for ticks, as well as any residing pathogens [99,100]. Although not associated with tularemia, the Gulf coast tick (*Amblyomma maculatum*) was reported to harbor a *Francisella*-like endosymbiont (FLE) that, despite undergoing genome reduction (i.e., one-third of coding genes, including virulence genes, were inactivated), gained the ability to synthesize cysteine, threonine and tyrosine. It is well-established that virulent *F. tularensis* strains are deficient in biosynthesis pathways for many amino acids, including cystine, threonine, and tyrosine [76], so it is interesting to speculate whether FLEs provide these amino acids to support *F. tularensis* infections in ticks.

Non-pathogenic *Coxiella* endosymbionts have been detected from *Amblyomma* sp., *Dermacentor* sp., *Hyalomma* sp., *Ixodes* sp., *Ornithodoros* sp., and *Rhipicephalus* sp. ticks. *Rickettsia* endosymbionts are common in *Ixodes* and some *Ornithodoros* tick species, while FLEs have been found in *Amblyomma*, *Dermacentor*, and *Hyalomma* ticks [31]. However, the number of different endosymbionts and prevalence of endosymbionts in each tick are known to vary by geographic location, time of year collected, life stage, sex, tick species, and lab-reared vs. wild-caught ticks [31,101,102,103,104]. A comprehensive study of over 80 tick species from around the world, including lab-reared and wild-caught ticks, identified the most prevalent tick endosymbiont to be a *Coxiella*-like endosymbiont (CLE) [31]. Although CLEs were thought to be evolutionarily stable in many tick species, further analysis showed that four other genera of endosymbionts, including *Francisella, Rickettsiella, Rickettsia*, and *Spiroplasma*, are slowly replacing CLEs as alternative obligate symbionts [31]. The presence of vitamin B2 and B7 synthesis pathways in FLEs, which are absent in CLEs, may explain why FLEs are becoming more common in several tick species, including *Amblyomma*, *Dermacentor*, and *Hyalomma* [31]. Although *F. tularensis* is known to infect these three tick genera, questions remain about how FLEs or other tick endosymbionts influence the ability of *F. tularensis* to infect, persist, and be transmitted by ticks [33,34,105].

To answer such questions, one group examined how the tick microbiome influenced *F. novicida* infections of *D. andersoni* ticks. In those studies, field-collected *D. andersoni* ticks contained a mixture of an unknown *Francisella* sp. and a FLE. Those endosymbionts were disrupted by feeding *D. andersoni* ticks on antibiotic-treated hosts. When those same ticks were experimentally-infected with *F. novicida*, the ticks harbored 0.5-log less *F. novicida* compared to non-antibiotic treated *D. andersoni* ticks. Those results suggested that FLEs and non-pathogenic *Francisella* endosymbionts may promote infection and persistence by other *Francisella* sp. [99]. Separate studies have shown that the midguts of *Amblyomma aureolatum* ticks, not known to transmit *F. tularensis*, are dominated by FLEs. Given the presence of FLEs in those ticks, but absence of *F. tularensis*, that study demonstrated that ticks harboring FLEs will not necessarily become infected with, or transmit, tularemia [106]. Taken together, much remains to be learned about the true role of non-pathogenic *Francisella* sp., FLEs, and other tick endosymbionts in ticks commonly-associated with tularemia (e.g., *D. variabilis* and *A. americanum*) in promoting infections by pathogenic *F. tularensis* strains.

On Martha’s Vineyard, and two other locations in Massachusetts, two distinct FLEs were reported in *D. variabilis* ticks. One FLE (designated as *D. variabilis* symbiont, DVS) was detected in 100% of ticks and a newly-identified FLE (designated as *D. variabilis francisella*, DVF; 96.6% similar to DVS) was detected in 55% of those same ticks. While DVS was restricted to ovarian tissues and Malpighian tubules, DVF was found in the hemolymph, suggesting that salivary gland colonization also was possible [41]. Given the history of *F. tularensis* infections on Martha’s Vineyard and high prevalence of DVS in these *D. variabilis* ticks [40,41], it is interesting to speculate about whether one FLE has recently evolved, whether one FLE is replacing the other, and what role each FLE plays in tularemia transmission.

Further studies examining FLE prevalence in ticks were conducted in southern Indiana, where two FLEs were identified in *D. variabilis* ticks. One FLE, accounting for over 75% of all identified microbes in *D. variabilis*, was present in 100% of ticks, and was identical to the DVS endosymbiont in *D. variabilis* ticks from Martha’s Vineyard [41,107]. This finding indicates that the DVS endosymbiont is not limited to Martha’s Vineyard and may colonize *D. variabilis* ticks across the country. A second FLE was detected in Indiana *D. variabilis* ticks, matching to the DVF endosymbiont in *D. variabilis* ticks from Martha’s Vineyard [41,107]. DVF accounted for approx. 10% of all tick microbiota sequences, suggesting that DVF is not an obligate endosymbiont. In Ontario, Canada, FLEs only were detected in 44% of *D. variabilis* ticks. However, those authors noted that their sequencing methods may not have been optimal for detecting low abundance organisms [108]. A more comprehensive study examined *D. variabilis* tick microbiomes from 26 U.S. states and five Canadian provinces, finding FLEs in only 24% of those ticks [109]. However, those authors also noted that a low abundance of endosymbiont DNA may have contributed to false negatives. Two phylogenetically-distinct FLE groups were identified in that study: one FLE group spanning the entire U.S. east of the Rockies (including most of southern Canada); and a second distinct FLE group present on the west coast of the U.S. Taken together, while much remains to be learned about the molecular mechanisms by which FLEs and endosymbionts support *F. tularensis* infections of *D. variabilis* (and other ticks), these studies do indicate that FLEs may play important roles in infection and transmission of *F. tularensis* by ticks [99].

Given that previous studies have demonstrated how differences in microbiomes alter tick competence for pathogen infections [99,110], more comprehensive studies are needed to examine tick microbiomes and understand how those microbiomes influence *F. tularensis* infections of ticks. Importantly, none of the above studies examined tick microbiomes from the south-central United States, where tularemia is most prevalent [28,109], nor did they examine *A. americanum* tick microbiomes. Additionally, is it unclear how identified non-pathogenic *Francisella* sp. and/or FLEs differ in their abilities to act as tick endosymbionts or to promote *F. tularensis* infections. Clearly, more studies are needed to isolate and sequence these endosymbionts to answer these questions.

There are important practical applications of arthropod microbiome studies, as highlighted by the use of the bacterium *Wolbachia* to colonize and compete for resources with Dengue virus in mosquitos [111]. Similarly, mechanisms that alter the microbiota of ticks or reduce obligate endosymbionts (e.g., DVS) from ticks may decrease *F. tularensis* acquisition or persistence in the environment. As highlighted above, more studies are needed to better understand which ticks are capable of being infected by, harboring, and transmitting *F. tularensis*, to study the diversity of endosymbionts that colonize those tick vectors, and to examine how those tick endosymbionts promote infection, persistence, and transmission of tularemia (Figure 1). Finally, although some *Rickettsia* endosymbionts have been shown to cause transient inflammation in animals and humans [112,113], it still is not known whether non-pathogenic *Francisella* sp. or FLEs are pathogenic to animals or humans.

## 8. The Tick Immune System and Antimicrobial Activities

The tick immune system has both cell-mediated and humoral defense mechanisms [114,115,116]. Hemocytes in the tick hemocoel are the primary cells of the tick cell-mediated immune system, recognizing and clearing microbes using processes including nodulation (aggregation), encapsulation, and phagocytosis. *D. variabilis* hemocytes have been shown to quickly aggregate around Gram-positive and Gram-negative bacteria inside the tick hemocoel, forming nodule-like masses, followed by hemocyte encapsulation around bacteria [117,118]. At least three different types of hemocytes exist in *D. variabilis*, namely plasmatocytes, type-1 granulocytes, and type-2 granulocytes, which form a multilayered capsule around foreign objects [119]. While studies have not been performed to examine aggregation or encapsulation of *F. tularensis* in *D. variabilis* or *A. americanum*, it is interesting to note that descriptions of tick hemocyte encapsulation of bacteria, including formation of a necrotic core in the center of these nodule-like masses, resemble descriptions of granuloma-like masses in *F. tularensis*-infected animals and humans [120,121,122]. *D. variabilis* hemocytes also have been shown to directly phagocytose and eliminate *B. burgdorferi* from infected ticks in <24 h [123]. Conversely, hemocyte phagocytosis does not necessarily kill all bacteria, as *A. phagocytophilum* infection of *I. scapularis* hemocytes is required for migration of the pathogen from the midgut to tick salivary glands [124]. Similarly, *R. parkeri*, a member of the spotted fever group, was detected in *A. americanum* tick hemocytes >1 month after infection, providing further evidence that tick hemocytes may not clear all bacterial infections and suggesting that some bacteria may evade tick humoral immune responses by infecting hemocytes [125]. Parallel comparisons have been made to help explain why *F. tularensis* infects mammalian macrophages—to evade humoral immune responses (e.g., antibodies) and other host immune responses [126,127]. Clearly, studies are needed to examine how *D. variabilis* and *A. americanum* (and potentially other ticks) hemocytes interact with *F. tularensis* and whether *F. tularensis* infects tick hemocytes to evade extracellular immune responses.

The tick humoral immune system includes extracellular effector molecules, such as lectins, complement-related molecules, and antimicrobial peptides [114,115,116]. The sequencing and annotation of the *I. scapularis* genome revealed a large number of putative genes that may protect ticks from microbial infections including defensins, lysozyme, peptidoglycan-recognizing proteins, and other putative antimicrobial peptides [128]. Defensins are small (4–6 kDa) cationic peptides that are active against Gram-negative and Gram-positive pathogens by attaching to and inserting in membranes, then forming pores [129]. *D. variabilis* has been shown to express a number of antimicrobial peptides, including defensin-1, defensin-2, lysozyme, and kunitz-type serine protease inhibitor, that limit infection by *R. montanensis*, another member of the spotted fever group [130,131,132]. Another *D. variabilis* defensin, varisin (5.3 kDa peptide), was reported to work together with lysozyme to lyse *B. burgdorferi* [133]. Conversely, depletion of varisin from *D. variabilis* ticks reduced *Anaplasma marginale* infections, suggesting that the true role of defensins in tick–pathogen interactions may be complex [134,135]. A recent proteomic analysis of *A. americanum* tick proteins produced during a blood meal indicates that this tick species produces at least six antimicrobial peptides, including kunitz-type serine proteinase inhibitor, lipochalin, microplusin, lysozyme, and defensins [136]. Many other potential tick immune molecules, such as fibrinogen-related proteins (FREPs) and thioester-containing proteins (TEPs), have been identified in soft and hard ticks and have been proposed to play important roles in protecting ticks from microbial infections [114,115,116]. However, the expression of these lesser-known molecules in either *D. variabilis* or *A. americanum* ticks has not been examined and their antibacterial effects on *F. tularensis* is unknown. In summary, although a great deal has been learned about the tick immune system, this field is rapidly evolving, and new mechanisms of tick cellular and humoral immunity are likely to be discovered at a rapid pace. Given the lack of previous studies to compare/contrast how *D. variabilis* and *A. americanum* immune systems respond to *F. tularensis* infections and recent expansion of high-throughput discovery approaches in ticks (e.g., genomics, proteomics, metabolomics) [115], future studies should examine how different tick vectors respond to *F. tularensis* infections. Such information could help explain why ticks such as *D. variabilis* and *A. americanum* harbor and transmit *F. tularensis*.

## 9. Conclusions

Despite a number of excellent studies that have examined *F. tularensis* (or *F. novicida*) in different tick vectors, we still do not understand: (1) how *F. tularensis* responds to the dramatic shift from mammals to ticks (e.g., temperature, iron concentration, and chitin); (2) how *F. tularensis* persists in ticks for months (e.g., Does *F. tularensis* infect/invade tick cells? Does *F. tularensis* migrate to/colonize salivary glands?); (3) how *F. tularensis* is transmitted from infected ticks to naïve hosts (e.g., Does *F. tularensis* actively sense and respond to a blood meal to promote transmission?); and (4) which tick(s) or arthropod vectors pose the greatest threat for transmitting tularemia to humans.

Although a wealth of information is available regarding *B. burgdorferi* genes/proteins that contribute to infection, persistence, and transmission by *I. scapularis* ticks, little is known about *F. tularensis* genes/proteins that confer similar functions. In fact, only one study has examined the role of *F. tularensis* genes in ticks. In those studies, *F. tularensis purMCD* was not found to be important for infection of ticks [34]. Based on previous studies that have identified specific genes/proteins required for a given tick-borne pathogen to infect its tick vector, similar or homologous *F. tularensis* genes/proteins would be obvious targets. However, given the unusual nature of the *F. tularensis* genome [137], other gene targeting strategies may be needed (e.g., *F. tularensis* transposon library screening in ticks; transcriptional analysis of *F. tularensis* in ticks). In addition, despite the availability of a number of tick cell lines and avirulent *F. tularensis* strains, future studies should examine how virulent *F. tularensis* strains successfully infect ticks, how *F. tularensis* survives the toxic process of blood digestion in a tick, how *F. tularensis* persists in ticks for months (i.e., overwintering) with very limited nutrients, how ticks respond to *F. tularensis* infections (e.g., tick antimicrobial responses), how *F. tularensis* responds to or actively modulates tick responses (e.g., bacterial immunomodulatory molecules), and how *F. tularensis* is transmitted to naïve hosts by infected ticks (including comparing transmission efficiency by different ticks, comparing host immune responses to *F. tularensis* infection by different ticks, and comparing disease progression after transmission by different ticks).

As outlined above and in Figure 1, there are many factors which may influence *F. tularensis* infections of ticks. To initially establish infections of ticks, *F. tularensis* may sense changes in temperature and iron levels. Although the tick blood meal is replete with nutrients, harmful breakdown products (e.g., ROS and RNS) and excess iron may damage *F. tularensis.* Tick immune responses and/or antimicrobial peptides may inhibit *F. tularensis* persistence and growth. Following blood meal processing, nutrients are likely extremely limited in the tick [138,139]. However, during tick molting, chitin fragments and components of the peritrophic matrix may be an important carbon source for *F. tularensis*. Some tick endosymbionts may provide nutrients to *F. tularensis* that aid in long-term persistence and replication. Conversely, other tick endosymbionts may compete with *F. tularensis* for nutrients or may stimulate tick antimicrobial defenses that restrict *F. tularensis* replication. All of these factors (Figure 1), and others, could impact the ability of *F. tularensis* to infect, persist, replicate, and be transmitted to naïve hosts. As such, these factors, and others not highlighted in this review, should be examined in future studies. All of the above highlighted studies are important, not only to better understand *F. tularensis* and to limit tularemia infections, but to provide information that is broadly-applicable to the tick-borne disease and vector-borne disease fields.

## Figures and Tables

**Figure 1 microorganisms-08-01639-f001:**
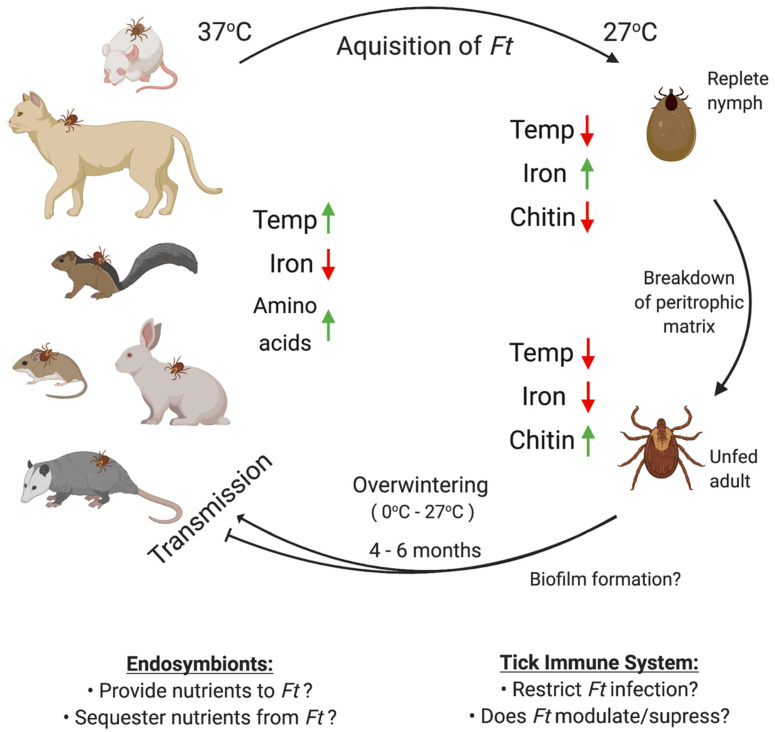
Possible factors affecting *F. tularensis* infection, persistence, and transmission in ticks. Ticks acquire *F. tularensis* (*Ft*) by taking a blood meal from an infected host (e.g., mouse, cat, squirrel, vole, rabbit, or opossum). Upon infecting ticks, *F. tularensis* likely regulates gene expression based on changes in temperature (37 to 27 °C) and iron (low to high). Following processing of the blood meal, nutrients are likely limited in the tick. However, during the tick molting process, chitin fragments may be made available via chitin remodeling and/or breakdown of the peritrophic matrix. Subsequently, *F. tularensis* must overwinter in ticks, a process which may require biofilm formation, evasion/modulation of the tick immune system, and/or interactions with tick endosymbionts. However, previous studies have not examined any of these factors. Upon transmission from infected ticks to a new mammalian host, *F. tularensis* likely senses mammalian cues (e.g., increased temperature, low iron, higher concentrations of amino acids) and may alter its gene expression, including virulence genes, to promote infection. Figure created using Biorender.com.

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
