# Peer review of "Mechanisms Affecting the Acquisition, Persistence and Transmission of *Francisella tularensis* in Ticks"

_microorganisms, 2020, doi:10.3390/microorganisms8111639_

Round 1
Reviewer 1 Report
The manuscript entitled “Mechanisms Affecting the Acquisition, Persistence and Transmission of Francisella tularensis in Ticks” and submitted as a review provides information on the what little is known about ticks as a vector for the spread of tularemia. In general, the manuscript is well-written, but many sentences are unnecessarily repeated throughout the review or the information is not relevant. Some statements are not supported by a reference and/or are inaccurate. Also, there is no detailed information provided on the tick immune system which is very relevant to this review, and there is no distinction made between subtype A1 and A2 F. tularensis strains which differ substantially in virulence throughout the manuscript. Therefore, this manuscript needs major revision before it is worthy of publishing in Microorganisms. The following lists the major and minor issues that need to be corrected or addressed.
1.) Abbreviations need to be defined in first use (e.g., United States).
2.) The statement on line 18 in the abstract is inaccurate. Please revise.
3.) The statement on line 48 should be explained in more detail as to the reasons why LVS is not licensed.
4.) The sentence on lines 53-54 is not accurate, since the causative agent in the majority of tularemia cases is unknown. Therefore, additional details are need to correct this statement.
5.) What is meant by “changes in precipitation” on line 59, such as increases or decreases?
6.) The wording on lines 93-95 is unclear and the following sentence of 96-98 is incomplete with many more states reporting cases of tularemia.
7.) The point of the sentence on lines 112 -115 is unclear. Please clarify or delete this sentence.
8.) Delete the word “total” in the sentence on line 122.
9.) Since there is no evidence that virulent Francisella tularensis strains produce biofilms, this text should be removed from the diagram in Figure 1.
10.) The sentence on lines 237-239 needs references to support this statement.
11.) F. tularensis needs to be italicized every time it is stated.
12.) Sentence on lines 270-272 needs more details, such as how many mice were used for each group and what type A and type B strains were used in the study.
13.) The phrase “to any appreciable degree” on line 287 should be deleted.
14.) The title and content of the section on “F. tularensis Biofilms” on lines 274-310 needs major revision. The relevance to the virulent F. tularensis strains in this section is very limited. F. novicida is only an opportunistic bacteria.
15.) The sentence on lines 318-320 is redundant, and the point in the following two sentences is unclear.
16.) The two sentences on line 331-336 need to be supported by references.
17.) The reference to Hyalomma ticks on line 339 needs more clarification.
18.) The sentences on lines 349-355 are unclear and need clarification.
19.) The sentences on lines 359-385 is not relevant to this review of F. tularensis and ticks. Please reword these sentences for relevance.
20.) The sentences on lines 406-412 are redundant and therefore, should be omitted.
21.) The sentence on lines 444-445 needs references to support this statement.
22.) The sentence on lines 447-448 is not relevant and should be deleted.
23.) Throughout the manuscript, there is no distinction made between subtype A1 and A2 F. tularensis strains which differ substantially in virulence. This needs to be corrected throughout the manuscript and the strains used in the studies discussed should be appropriately provided.
24.) There is no detailed information provided on the tick immune system which is very relevant to this review. More information on this topic needs to be included.
Author Response
1. Abbreviations need to be defined in first use (e.g., United States).
Author response: Abbreviations including United States (lines 11 and 27) and lipopolysaccharide (line 246) have been defined in their first use.
2. The statement on line 18 in the abstract is inaccurate. Please revise.
Author response: The previous sentence on line 18, about foci of infections in the south-central United States, has been removed.
3. The statement on line 48 should be explained in more detail as to the reasons why LVS is not licensed.
Author response: As suggested, we have added more information on why LVS is not licensed in lines 56-58
4. The sentence on lines 53-54 is not accurate, since the causative agent in the majority of tularemia cases is unknown. Therefore, additional details are needed to correct this statement.
Author response: We have revised this statement, line 64, to note that approximately half of U.S. tularemia cases are associated with tick bites.
5. What is meant by “changes in precipitation” on line 59, such as increases or decreases?
Author response: In line 69, we have changed “changes in precipitation” to “increased precipitation”, as indicated in the publication.
6. The wording on lines 93-95 is unclear and the following sentence of 96-98 is incomplete with many more states reporting cases of tularemia.
Author response: We agree with the Reviewer’s comments and, in response to a similar comment from Reviewer #3, have deleted this paragraph in the revised manuscript. The deleted paragraph followed line 103.
7. The point of the sentence on lines 112 -115 is unclear. Please clarify or delete this sentence.
Author response: Lines 140-145 in the revised manuscript have clarified previous publications on D. andersoni ticks and have clarified that more information is needed about tick species that transmit tularemia to humans.
8. Delete the word “total” in the sentence on line 122.
Author response: The word “total” has been removed from line 151 (previously “with 1 total fatality”)
9. Since there is no evidence that virulent Francisella tularensis strains produce biofilms, this text should be removed from the diagram in Figure 1.
Author response: We appreciate the Reviewer’s comments and agree that there is no evidence that virulent strains of F. tularensis form biofilms to persist/survive in ticks. For this reason, a question mark followed “Biofilm formation?” in Figure 1 and the Figure 1 legend specifically notes “F. tularensis must overwinter in ticks, a process that may require biofilm formation…” We respectfully remind Reviewer #1 that this is a review article and, after highlighting relevant studies (Section 5, lines 349-364), our goal was to ask reasonable questions that promote future research studies. We carefully considered previous studies on F. tularensis biofilms (Section 5, lines 349-365). Notably, results from Champion et al. 2019 (reference 96) and Margolis et al. 2010 (reference 93), both demonstrated that virulent Type A and Type B F. tularensis strains do, in fact, form biofilms, albeit more slowly and less dense than F. novicida. Our deliberate choice of the words “may require” (Figure 1 legend) and use of a question mark (Figure 1) highlight that it remains to be determined if virulent F. tularensis strains form biofilms in ticks. We have clarified this point in the Biofilm section (lines 362-365), now noting that “future studies should examine biofilm formation in ticks by virulent F. tularensis strains.”
10. The sentence on lines 237-239 needs references to support this statement.
Author response: As suggested, Reif et al 2018 (reference 5) has been added after line 311 in the revised manuscript.
11. F. tularensisneeds to be italicized every time it is stated.
Author response: We apologize for these errors. F. tularensis has been italicized throughout the manuscript, including previous omissions in line 342 and line 348.
12. Sentence on lines 270-272 needs more details, such as how many mice were used for each group and what type A and type B strains were used in the study.
Author response: The relevant strains and sample sizes have been added to lines 344-345.
13. The phrase “to any appreciable degree” on line 287 should be deleted.
Author response: The phrase “to any appreciable degree” has been removed form line 361 of the updated manuscript. This is in reference to Type A and Type B biofilm formation, which has been revised. See Author response #9, above.
14. The title and content of the section on “F. tularensis Biofilms” on lines 274-310 needs major revision. The relevance to the virulent F. tularensisstrains in this section is very limited. F. novicidais only an opportunistic bacteria.
Author response: As noted in author responses #9 and #13, the biofilm section has been revised. We have clarified differences in F. novicida (opportunistic pathogen) and the ability of virulent F. tularensis strains to form biofilms. This revised section is lines 349-365. In addition, for clarity, we have created a new section, entitled “F. tularensis Chitin Utilization”, lines 367-398, to separate previous studies on biofilms with molecular characterization of chitinases.
15. The sentence on lines 318-320 is redundant, and the point in the following two sentences is unclear.
Author response: The first paragraph of the Tick Endosymbiont section was revised for clarify to avoid redundancy. Previous lines 318-320 were removed, as suggested. Lines 410-413 have been added to clarify the potential role of FLEs in providing amino acids to virulent F. tularensis.
16. The two sentences on line 331-336 need to be supported by references.
Author response: Lines 419-24 are referenced by Duron et al. 2017 (Reference 31)
17. The reference to Hyalomma ticks on line 339 needs more clarification.
Author response: We agree that the note about Hyalomma ticks was confusing in the previous manuscript. This has been removed (line 427-429)
18. The sentences on lines 349-355 are unclear and need clarification.
Author response: This is in reference to the study that examined D. andersoni endosymbionts, antibiotic treatment of those ticks, and subsequent F. novicida infection. This section has been majorly revised. Lines 449 to 461.
19. The sentences on lines 359-385 is not relevant to this review of F. tularensisand ticks. Please reword these sentences for relevance.
Author response: We respectfully disagree with Reviewer #1. Section 7 is focused on Tick Endosymbionts, including FLEs (and other non-pathogenic Francisella endosymbionts) and what has been published on D. variabilis endosymbionts. D. variabilis is arguably the most important tick vector for tularemia. As such, it is important to review papers that have examined D. variabilisendosymbionts across the U.S. and Canada. We acknowledge that relevance was lacking in the original submission. This section, lines 399-610 has been extensively revised and edited to clarify relevance and future directions.
20. The sentences on lines 406-412 are redundant and therefore, should be omitted.
Author response: As suggested, we have deleted lines the previous lines 406-412 from the conclusion. Please see the revised Conclusion, beginning with line 845.
21. The sentence on lines 444-445 needs references to support this statement.
Author response: This is in reference to nutrients being extremely limited in the tick. As suggested, two references have been added to line 893 of the revised manuscript.
22. The sentence on lines 447-448 is not relevant and should be deleted.
Author response: This is in reference to F. tularensis forming biofilms in ticks. As suggested, we have deleted the sentence from the conclusion.
23. Throughout the manuscript, there is no distinction made between subtype A1 and A2 F. tularensisstrains which differ substantially in virulence. This needs to be corrected throughout the manuscript and the strains used in the studies discussed should be appropriately provided.
Author response: Additional information regarding F. tularensis subtypes A1a, A1b, A2, and Type B, and strains used in previous studies have been added throughout the revised manuscript, including lines 45-56, 96, and 343-344.
24. There is no detailed information provided on the tick immune system which is very relevant to this review. More information on this topic needs to be included.
Author response: A new section on the tick immune system has been added to the revised manuscript, lines 631-823. We believe that this new section provides substantial information that enhances the review.
Reviewer 2 Report
General comments
The manuscript of Tully and Huntley represents an up-to-date review about the facts known about the Francisella - tick interaction. This is a very well written review with interesting facts for the reader and the manuscript exhibits a high amount of very recent publications.
I have only some minor comments:
- The numbering of the different pragraphs is missing
- lines 52 and 114: I would prefere to describe F. novicida as an opportunistic pathogen, able to infect immunocompromised humans, but not as "not virulent/avirulent in humans", in my opinion this is not realy correct.
- line 63: In Zellner and Huntley Poland is mentioned to be part of the other group, and there are two references for this (Wojcik-Fatla et al., 2015; Bielawska-Drozd et al., 2018).
- lines 86-88: Is there then the opportunity of ticks harboring two different Francisella strains?
- line 159: please delete
- line 235: delete "Incection", is stated in line 158
- line 257-273: You may mention "transstadial transmission" in this section and cross reference to "Zellner and Huntley 2019"
- line 391: You may mention some examples (names) of the non-pathogenic Francisella sp. "you mean here".
- In my opinion the conlusion is a little bit too long and may be shortened.
Author Response
1. The numbering of the different paragraphs is missing
Author response: Paragraph/section numbering has been added.
2. lines 52 and 114: I would prefer to describe F. novicida as an opportunistic pathogen, able to infect immunocompromised humans, but not as "not virulent/avirulent in humans", in my opinion this is not really correct.
Author response: The description of F. novicida has been changed to “rarely associated with disease in immunocompromised humans” on line 60-61 and “rare infection of immunocompromised humans” on line 1412.
3. line 63: In Zellner and Huntley Poland is mentioned to be part of the other group, and there are two references for this (Wojcik-Fatla et al., 2015; Bielawska-Drozd et al., 2018).
Author response: Two papers referencing F. tularensis-infected ticks in Poland were added to line 71.
4. lines 86-88: Is there then the opportunity of ticks harboring two different Francisella strains?
Author response: We are not aware of any publication assessing the ability of ticks to harbor multiple Francisella strains. No changes have been made to the manuscript since no publications exist and this is pure speculation.
5. line 159: please delete
Author response: The previous sub-section header “Temperature, pH and Iron” has been deleted from line 229-230.
6. line 235: delete "Infection", is stated in line 158
Author response: The previous section title “F. tularensis Infection, Persistence and Transmission in Ticks” has been changed to “F. tularensis Persistence and Transmission in ticks” on line 307.
7. line 257-273: You may mention "transstadial transmission" in this section and cross reference to "Zellner and Huntley 2019"
Author response: As suggested, transstadial transmission was added to line 341 and the “Zellner and Huntley, 2019” reference (reference 12).
8. line 391: You may mention some examples (names) of the non-pathogenic Francisella sp. "you mean here". Author response: The names of various non-pathogenic Francisella sp. and/or endosymbionts, discussed in line 617, are listed throughout the Tick Endosymbiont section (section 7), lines 401-630. To avoid redundancy, those names are not listed again on line 617. Please also see author response #19 to Review #1, where we noted that the Tick Endosymbiont section has been extensively revised and edited to clarify relevance and suggest future research studies.
9. In my opinion the conclusion is a little bit too long and may be shortened.
Author response: Reviewer #1 also commented that portions of the conclusions were redundant. See response #22 to Reviewer #1. The revised conclusion section (Section 9) has been shortened, including deleting the first paragraph, which was redundant.
Reviewer 3 Report
I have an opportunity to read the MS entitled “Mechanisms Affecting the Acquisition, Persistence and Transmission of Francisella tularensis in Ticks” by Huntley and Tully.
The authors have published a nice review “Ticks and Tularemia: Do We Know What We Don't Know, Front. Cell. Infect. Microbiol. 2019”. Some parts of the description are overlapped, especially introduction section. Please rearrange introduction to relate the main subject of this paper.
Line 174-182. In current form, the description should be removed because there is not enough scientific evidence to mention in the review. Do the authors know any evidences the relationship of pH and F. tularensis (ie, grow in culture). If yes, please described in this part.
Author Response
1. The authors have published a nice review “Ticks and Tularemia: Do We Know What We Don't Know, Front. Cell. Infect. Microbiol. 2019”. Some parts of the description are overlapped, especially introduction section. Please rearrange introduction to relate the main subject of this paper.
Author response: We appreciate the reviewer’s comments. We carefully reviewed our previous publication in Front. Cell Infect. Microbiol, made a number of revisions to the current manuscript (including deleting a paragraph from the introduction, rearranging other sections, and adding a new section on the tick immune system). The revised Introduction includes many changes, additions, and clarifications. Please see responses to Reviewers #1 and #2 about changes to the introduction. To avoid overlap, previous lines 93-99 have been removed (tularemia cases in individual U.S. states; tularemia in various animals, etc.). The final paragraph of the introduction has been revised to outline topics that will be highlighted in this review.
2. Line 174-182. In current form, the description should be removed because there is not enough scientific evidence to mention in the review. Do the authors know any evidences the relationship of pH and F. tularensis (ie, grow in culture). If yes, please described in this part.
Author response: We understand the reviewer’s concern. As noted in our response to Reviewer #1, this is a review article and our goal was to highlight gaps in information and suggest potential topics that could be explored in future research studies. Since B. burgdorferihas been shown to adapt to life in the tick by recognizing changes in environmental cues (including pH changes), it is possible that F. tularensis also may sense these changes. In the revised manuscript, lines 244-252, we have added a reference that Type A and Type B (LVS) F. tularensis are resistant to acid stress and are viable at pH 3 (line 249). Despite previous research studies showing that F. tularensis regulates virulence genes based on pH changes in the mammalian phagosome, we are not aware of any previous publications examining pH-regulated F. tularensis gene changes in the tick.
Round 2
Reviewer 1 Report
The manuscript entitled “Mechanisms Affecting the Acquisition, Persistence and Transmission of Francisella tularensis in Ticks” and submitted as a review provides information on the what little is known about ticks as a vector for the spread of tularemia. The revised manuscript is now acceptable for publication in Microorganisms.
Reviewer 3 Report
I do not have any concerns in the revised MS.